# An automated morphometric approach to evaluate distal lung patterning in mouse models of bronchopulmonary dysplasia

Declan J. Gainer[1], Mark L. Ormiston [1,2,3]*

**1** Queen's University Department of Medicine, Kingston, Canada, **2** Queen's University Department of Biomedical and Molecular Sciences, Kingston, Canada, **3** Queen's University Department of Surgery, Kingston, Canada

\* mark.ormiston@queensu.ca

## Abstract

### Background

Chronic respiratory diseases represent a large group of non-communicable diseases that are a leading cause of mortality and morbidity globally. Many of the methods utilized to assess airway simplification in experimental models of the conditions are overly time-consuming and are sensitive to inter-operator biases, necessitating the need for unbiased and efficient tools to supplement analyses.

### Methods

We propose a semi-automated method to quantitate the characteristics of large terminal respiratory airways and alveoli that uses free image-processing software (Fiji). We aimed to develop and test this method in a mouse model of bronchopulmonary dysplasia (BPD), a disease of blunted airway and pulmonary vascular development that remains a leading cause of mortality among preterm infants. Optimal macro parameters were determined with a test set of images from postnatal day 14 (P14) mice exposed to acute postnatal hyperoxia by determining which area and circularity values best correlated with mean linear intercept ($L_M$). Validation was performed on a separate set of images from P7 mice subjected to the same hyperoxic model of BPD.

### Results

Both alveolar duct (r: 0.7866, p = 0.0359) and alveolar (r: 0.9475, p = 0.0012) area correlated with $L_M$ measurements from the test set. Using our method on a validation dataset, we demonstrate that hyperoxia-exposed mice possess fewer, enlarged alveoli that occupy less total area, as well as enlarged alveolar ducts that occupy a greater proportion of the parenchyma.

**Data availability statement:** All relevant data are within the manuscript and its Supporting Information files.

**Funding:** This work was supported by funding from the Canadian Institutes of Health Research (PJT_180356) awarded to M.L.O. The funding institution did not play a role in study design, data collection, analysis, decision to publish, or manuscript preparation.

**Competing interests:** The authors have declared that no competing interests exist.

## Conclusions

We report a semi-automated method of quantitating the characteristics of large and small terminal respiratory airways. This tool expedites analysis and removes operator bias relative to existing methods. We also demonstrate that $L_M$ changes in an acute model of hyperoxia-induced BPD result from both alveolar simplification and inadequate primary septation at the level of the alveolar ducts.

## Introduction

Many disorders are known to negatively influence the lung parenchyma and airways. Collectively, these disorders are known as chronic respiratory diseases (CRDs) [1], a group of non-communicable diseases that remain a leading cause of morbidity and mortality globally [2]. CRDs encompass both obstructive and restrictive lung diseases, which are known to manifest differing structural phenotypes as a consequence of underlying disease pathophysiology and environmental influences. Classically, obstructive diseases are characterized by the presence of emphysema-like structural changes, or the enlargement of distal airspaces due to the destruction of parenchymal tissue [3]. This enlargement can affect entire pulmonary lobules (panlobular), or can be primarily localized to the respiratory bronchioles (centrilobular) [4,5]. In comparison, restrictive diseases are driven by the accumulation and deposition of excess fibrotic tissue throughout the lung, in addition to matrix-related proteins such as elastin and collagen [6]. These structural changes not only impede gas exchange across the respiratory membrane, but also thicken airway walls, reducing their compliance and functional diameter [6].

Although lung diseases are generally classified as either obstructive or restrictive in nature, bronchopulmonary dysplasia (BPD) can exhibit characteristics of both, making it a phenotypically diverse disease [7]. BPD is a chronic lung disease of prematurity that has been linked to the blunted development of both the pulmonary circulation and airways, resulting in persistent airway dysfunction and an increased risk of various cardiopulmonary complications later in life [8]. This disease has been associated with a range of antenatal and postnatal risk factors, including pre-term birth and early-life respiratory support, and remains a leading cause of mortality and morbidity in preterm infants for which no curative treatments are clinically available. Rather, current therapies focus primarily on minimizing lung damage. Consequentially, experimental approaches to treat BPD are often tested in animal models, commonly mouse models of BPD, which typically involve postnatal exposure to hyperoxia (75%−95% $O_2$) for an extended duration, ranging from 3–14 days [9]. Models utilizing invasive ventilation, mixed hyperoxia and hypoxia, chorioamnionitis, and transgenic modifications have also be utilized.

Although experimental timelines and exposures are variable, continuous exposure to hyperoxia is known to blunt lung development during the late saccular and early alveolar stages of lung development. Much like in humans, this results in alveolar simplification, fibrosis, and the development of a dysregulated pulmonary vasculature

[9]. One of the challenges of these models are the methods used to quantify the impact of disease on airway patterning. Several methods are conventionally used to quantify these impacts, including radial alveolar counts (RAC) [10], mean linear intercept ($L_M$) [11], and gold-standard lung stereology [12]. These methods aim to provide quantitative measures of structural aspects of the distal lung, including airway size ($L_M$), alveolar septation (RAC), and absolute alveolar number and surface area (stereology). Although robust and commonplace, all three methods are time-consuming, particularly stereology, due to specific requirements for inflation, embedding, sectioning, and imaging. Additionally, operator bias and a lack of specificity for specific generations of airways are considerations, highlighting the need for time-efficient, unbiased tools that can quantitate the characteristics of different classes of airways. Here, we propose a tool to supplement these conventional measures in assessing disease in a mouse model of BPD that is capable of providing both airway counts and area measurements in a manner that reduces assessment time and circumvents operator bias.

## Materials & methods

### Ethics statement

All animal work was performed at Queen's University in Kingston, Ontario, Canada. Experimental protocols (#2024–2223) were approved by the University Animal Care Committee (UACC) and adhered to guidelines established by the Canadian Council on Animal Care (CCAC). Mice were housed in individually ventilated cages and received *ad libitum* access to food and water.

### Mouse model of bronchopulmonary dysplasia

Timed-pregnant C57BL/6 dams (Charles River Laboratories) were kept at room temperature on 12 h sleep/wake cycles until birth. Following birth on postnatal day 0 (P0), all newborn pups were assigned to (i) normoxia (21% $O_2$) or (ii) hyperoxia (95% $O_2$) groups. Pups in group (ii) and their respective dams were exposed to hyperoxia in a single-latch A-Chamber (BioSpherix) for 3 days [13], while pups in group (i) were housed in the same room under normoxic conditions. Dams were rotated pairwise between conditions every 24 hours to prevent oxygen toxicity and preserve maternal health. Pups were removed from 95% $O_2$ after 72 hours on P3, and were allowed to recover under normoxic conditions until either P7 or P14. All P7 mice received a 5 µL subcutaneous (S.Q.) injection of sterile PBS in the interscapular region prior to hyperoxic exposure. Animals were euthanized via an intraperitoneal overdose of Sodium Pentobarbital (100 mg/kg), in accordance with CCAC guidelines.

### Lung tissue harvest and processing

Following sacrifice, the chest cavity was opened, and the right ventricle was perfused with heparinized saline (1000 U/mL diluted 1:50 in phosphate buffered saline (PBS)). The lungs were inflated with 2% paraformaldehyde (PFA) at a pressure of 25 mmHg, removed *en-bloc*, and fixed in 2% PFA at 4°C for 48 hours in the dark. Lungs were subsequently washed overnight in 1X PBS at 4°C, embedded in paraffin, sectioned (5 µm), and stained with hematoxylin and eosin (Abcam). Slides were imaged using a Leica MICA (Leica Microsystems). Six (6) 20X high-powered fields of view (HPF) were taken per animal.

### Manual mean linear intercept scoring

$L_M$ scores were obtained from Fiji by overlaying a line of known length over a region of distal lung tissue, avoiding terminal respiratory bronchioles or vascular structures, and counting the number of intersections with respiratory airway walls. Scores were calculated as:

$$L_M = \frac{\overleftrightarrow{AB}}{N_{intersection}}$$

where $\overset{\leftrightarrow}{AB}$ is the length of the overlaid line, and $N_{intersection}$ is the number of times a given line intersected with a respiratory airway wall. Two measurements were taken per HPF by a single blinded observer, totalling 12 measurements per animal, which were averaged to compute the final $L_M$ score.

## Method workflow

Briefly, a Fiji [14] macro was written to provide a semi-automated method of quantitating the characteristics of distal airways, specifically alveolar ducts and alveoli. Obtained measurements, including tissue area, airspace area (individual and total), number, and percent of total area, are directly written to an excel file using an existing excel extensions repository [15]. Disconnected vascular exudates and small holes ($<50$ µm$^2$) are automatically removed and filled during image processing. Connected exudates, terminal respiratory bronchioles, and major vascular structures must be manually selected and deleted to prevent inclusion.

A user-defined scale must be set globally prior to use, to prevent inaccuracies in output values. Images of most formats and sizes will be accepted, though the macro was developed and tested using .tif image files. Inputs are processed and saved to an output directory for post-processing visual inspection, whereby deleted structures are made orange and highlighted airways are cyan in order to improve visual contrast.

The workflow for the method applies to all images being processed sequentially. Following the definition of input and output directories for both test images and the results datasheet, a calibration image is opened from the defined input directory. Here, a user-defined threshold can be set. This threshold assumes that all images to be processed in a batch are equivalently stained. Opened 24-bit RGB images are then converted to 8-bit greyscale by isolating the green channel (Fig 1A,B). Green was selected to provide superior contrast between stained tissues and the background [16]. Users are then prompted to manually select and remove exudates, vascular structures, respiratory bronchioles, pleural spaces, and other unwanted regions from analysis using the wand tool (Fig 1C). Images are then automatically thresholded using user-defined values from calibration (Fig 1D). Disconnected exudates, debris, and particulates are removed to minimize inaccuracies (Fig 1E). The "open" binary operator is applied to all images. Briefly, erosion and dilation are performed in succession to remove isolated pixels and smooth the image (Fig 1F). Small holes ($<50$ µm$^2$) in the lung parenchyma are removed to improve visualization (Fig 1G). This operation does not impact the resulting measurements, as all quantified structures are greater than this size. The characteristics of alveolar ducts and alveoli are quantified (Fig 1H,I), and results are written to an excel file in the specified output directory. All output images are automatically saved for visual inspection.

## Method optimization

The irregularity in both shape and size of terminal respiratory airspaces necessitated optimization of the particle analyzer's parameters. As $L_M$ is a measurement that is partially dependent on airspace size, we sought to identify the parameters that best correlated with manually-derived $L_M$ scores by evaluating several combinations of area and circularity values (Tables 1 and 2) using a dataset consisting of lung images from 7 animals. Animals from both healthy and diseased groups were included to ensure a range of $L_M$ values were represented. In this context, area refers to target airspace area, and circularity refers to the circularity of target airways.

As alveolar ducts are a site of primary septation they should, in principle, not be perfectly circular. For each alveolar duct airspace value, two circularity values were evaluated to determine whether the exclusion of highly circular structures improved quantitation (Table 1). Alveoli do not undergo further septation and, therefore, should trend towards higher circularity values. As such, three circularity values were evaluated for alveolar area values to determine whether the exclusion of highly noncircular airspaces would improve quantitation (Table 2).

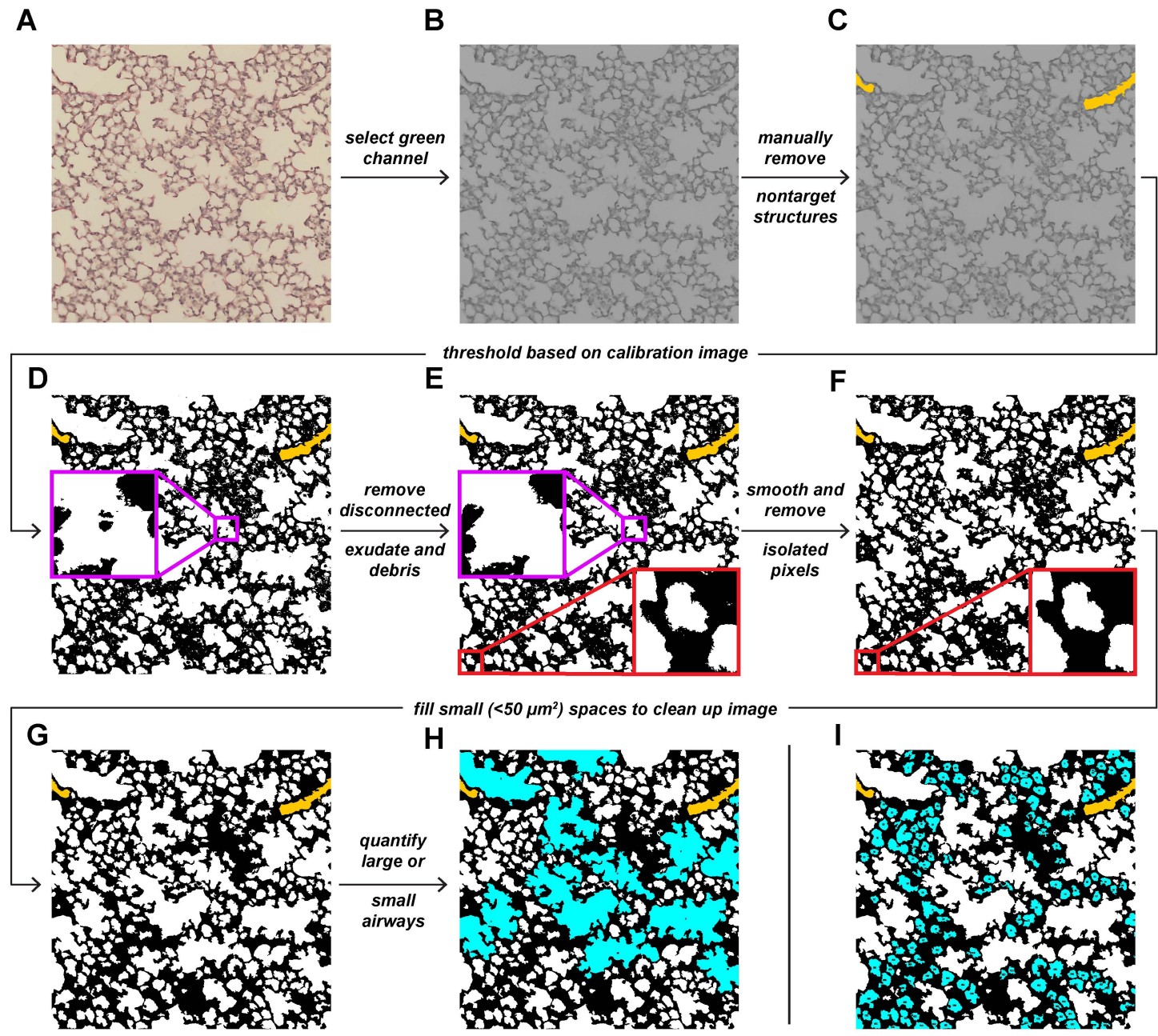

**Fig 1. Summary of image processing and analysis steps for quantifying alveolar duct and alveolar area.** (A) Representative RGB color image acquired at 20x magnification. (B) Isolated green channel of A. (C) Vessels removed from B. (D) Binarized version of C. (E) Disconnected exudate and debris removed from D. (F) Smoothed version of E. (G) Small holes removed from F. Selective quantification of (H) alveolar ducts or (I) alveoli.

**Table 1. Base area and circularity values tested for the quantification of alveolar ducts.**

| Airspace Area | 3000 μm² | | 4000 μm² | | 5000 μm² | | 6000 μm² | |
|---|---|---|---|---|---|---|---|---|
| Circularity values tested | 0-0.5 | 0-1 | 0-0.5 | 0-1 | 0-0.5 | 0-1 | 0-0.5 | 0-1 |

**Table 2. Base area and circularity values tested for the quantification of alveoli.**

| Base Area | 50 µm² | | | 100 µm² | | | 150 µm² | | |
|---|---|---|---|---|---|---|---|---|---|
| Circularity values tested | 0.01-1 | 0.1-1 | 0.25-1 | 0.01-1 | 0.1-1 | 0.25-1 | 0.01-1 | 0.1-1 | 0.25-1 |

The macro was applied to the same test images that underwent manual $L_M$ quantification using the above parameter combinations. To determine the optimal combination for each respiratory airway type, mean duct area and alveolar area measurements from each animal in the optimization dataset were correlated with $L_M$ values using an adapted leave-one-out (LOO) approach [17], reporting Pearson's r as the evaluation metric. A LOO approach was applied to optimization correlative analyses to reduce the influence of any single observation on reported r values, enabling more robust assessment of association in the context of a small $n$. Briefly, each sample from the $n=7$ optimization dataset was iteratively excluded, and Pearson's correlation between airway area and $L_M$ was computed on the remaining data ($n=6$). The median r coefficient across all iterations was compared to the r coefficient derived from the analysis using all 7 samples to determine which parameter combination most strongly and consistently correlated with $L_M$. As $L_M$ and area report values on different unit scales, both measurements were standardized using z-scoring to enable the assessment of proportional, but not fixed, bias and relative agreement by Bland-Altman analysis:

$$z = \frac{(x - \mu)}{\sigma}$$

Where x denotes the observed value, µ denotes the variable mean, and σ is the standard deviation of the variable.

## Method validation

Following the determination of optimal circularity and area values, the macro was tested on a separate test set of images from 22 animals. Outputs were compared to manually computed $L_M$ scores to determine whether the two measures correlated. Proportional bias and relative agreement were assessed by constructing Bland-Altman plots of standardized measurements, as described above.

## Statistical analysis

All data were collated in GraphPad Prism 10 (v10.6), which was used for statistical analysis. All data are presented as mean±SD. Assessment of significance between two groups was performed by unpaired, two-tailed Student's *t*-test. An *f*-test was used to assess differences in variance between experimental groups. If variances differed significantly, Welch's correction was applied. Pearson's correlation was used to assess the relationship between $L_M$ and alveolar duct/alveolar area. Prior to computing Pearson's r, normality was assessed for all data using a Shapiro-Wilk test. Linear relationships between $L_M$ and duct or alveolar area were confirmed with a scatterplot.

## Results

### Method optimization

To explore the potential for a macro to quantitate the characteristics of distal airspaces, we exposed newborn C57BL/6 mice to 95% $O_2$ as an acute model of hyperoxia-induced BPD. This mouse model of BPD arrests lung development as a consequence of hyperoxic exposure, leading to terminal airspace enlargement that can be visualized by histology. In keeping with literature, terminal airspace enlargement was apparent in P14 mice exposed to 95% $O_2$ for 72 hours at birth, relative to 21% $O_2$ controls, as quantified by $L_M$ (Fig 2A,B). To develop a method capable of quantitating the characteristics of distal respiratory airways, and differentiating between alveolar ducts and alveoli, a set of minimum area and circularity

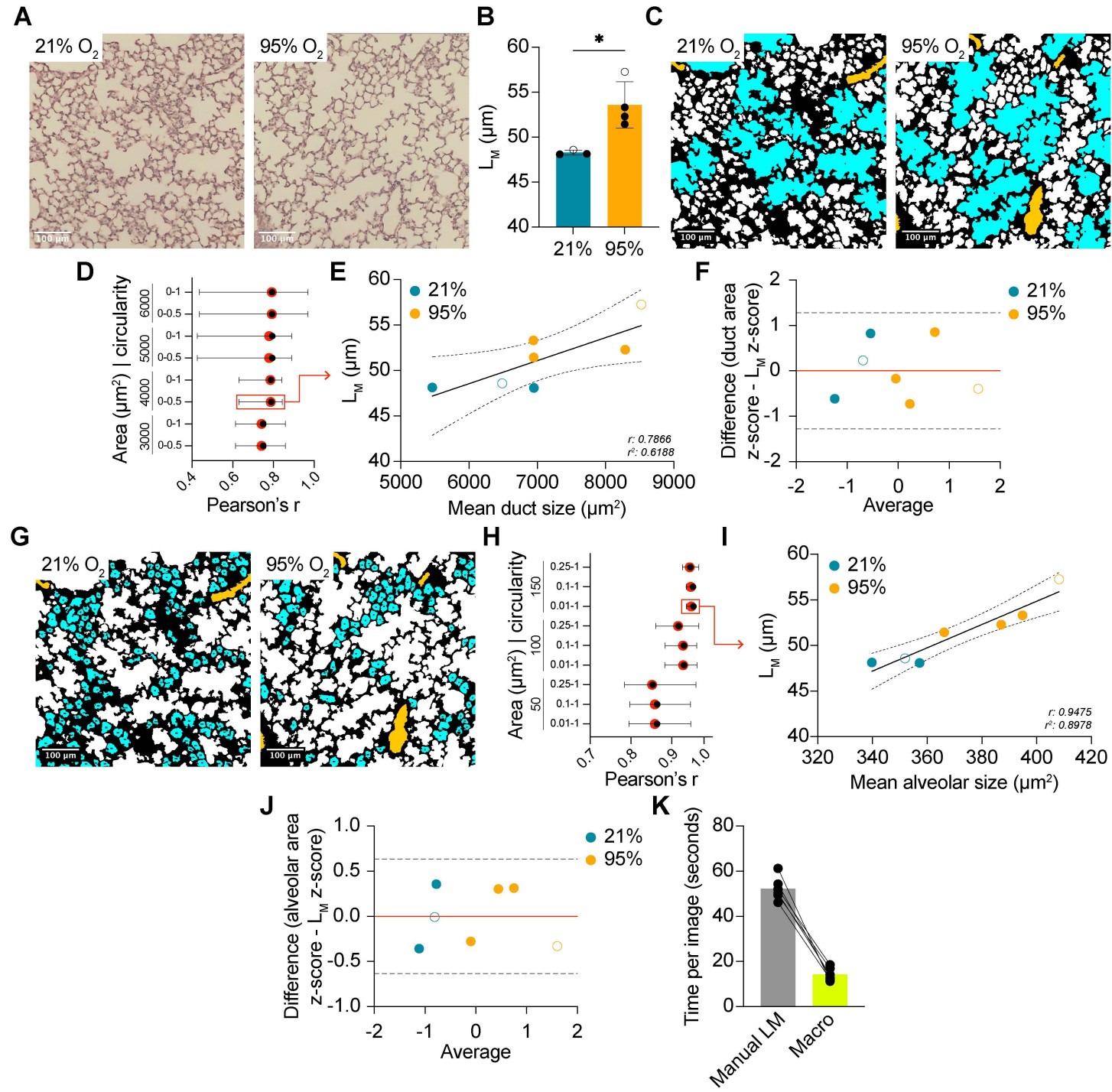

**Fig 2. Optimizing method parameters on images from mice exposed to an acute hyperoxia-induced model of BPD.** (A) Representative H&E-stained lung images from P14 mice exposed to 21% $O_2$ ($n=3$) or 95% $O_2$ ($n=4$) for 72 hours. (B) Quantification of mean linear intercept ($L_M$) in the mice from A. (C) Representative colored images depicting alveolar ducts. (D) Evaluation of Pearson's r across minimum area and circularity combinations for alveolar ducts. Red dots depict the r coefficient for all 7 tested samples. Black dots depict the median r coefficient for all 7 LOO iterations. Bars represent the range of values across all 7 LOO iterations. (E) Scatter plot of mean duct area plotted against measured $L_M$. Solid black line depicts the fitted linear regression line. Dashed lines represent the 95% confidence intervals. (F) Bland-Altman plot of z-scores for alveolar duct area and $L_M$. Dashed lines represent the 95% limit of agreement (LOA). Red line depicts bias. (G) Representative colored images depicting alveoli. (H) Evaluation of Pearson's r

across minimum area and circularity combinations for alveoli. Red dots depict the r coefficient for all 7 tested samples. Black dots depict the median r coefficient for all 7 LOO iterations. Bars represent the range of values across all 7 LOO iterations. (I) Scatter plot of mean alveolar area plotted against measured $L_M$. Solid black line depicts the fitted linear regression line. Dashed lines represent the 95% confidence intervals. (J) Bland-Altman plot of z-scores for alveolar area and $L_M$. Dashed lines represent the 95% limit of agreement (LOA). Red line depicts bias. (K) Bar graph depicting processing time per image across a test set of 6 images (1 animal). (○) female mice, (●) male mice. *$P < 0.05$. Student's t-test used in B. Simple linear regression used in E, I. Pearson's correlation used in D,H. Bland-Altman analysis used in F,J. Error bars are mean ± SD.

values to identify either alveolar ducts or alveoli were selected. A positive association between measured alveolar duct area and $L_M$ was observed across all tested parameter combinations (r: 0.7413–0.7922; Fig 2C,D). However, examination of the median and range of r values for the LOO iterations revealed that selecting for the largest alveolar ducts led to less stable associations with $L_M$. As such, 4000 µm² − 0-0.5 was selected as the parameter combination that strongly and consistently correlated with $L_M$ (r: 0.7866, p = 0.0359; Fig 2E). Bland-Altman analysis of duct area and $L_M$ z scores revealed no observations outside the 95% limits of agreement (LOA; 95% LOA: −1.280, 1.280), and no apparent proportional bias (Fig 2F). Similarly, a positive association was observed between measured alveolar area and $L_M$ across the tested parameter combinations (r: 0.8516–0.9465; Fig 2G,H). Although the median and range of r values across LOO iterations was consistently higher for alveolar area relative to duct area, 150 µm² - 0.01−1 correlated most stably and strongly with $L_M$ (r: 0.9475, p = 0.0012; Fig 2I). Bland-Altman analysis revealed no proportional bias, and no observations fell outside the 95% LOA (−0.6350, 0.6350). When compared to manual quantification of $L_M$, our method performed significantly (~3.5X) more efficiently, allowing for substantial time savings during batch processing (Fig 2K).

## Method validation

To assess the efficacy of the selected area and circularity parameters in quantitating the characteristics of alveolar ducts and alveoli, the optimized macro was applied to lung images from neonatal mice at P7, following exposure to the same model of hyperoxia-induced BPD. Although mice from the validation set were at an earlier stage of alveolar lung development than those from the optimization set, acute exposure to 95% $O_2$ manifested a similar increase in $L_M$ versus controls at P7 (Fig 3A,B). Consistent with this finding and previous studies [18], diseased mice had comparatively enlarged alveolar ducts that were more numerous and occupied a greater total area (Fig 3D,E,F). In keeping with previous findings, there was a strong positive relationship between mean alveolar duct area and $L_M$ (r: 0.7651, p < 0.0001; Fig 3G) and no apparent proportional bias (Fig 3H). However, one sample from the BPD group fell outside the limits of agreement (95% LOA: −1.344, 1.344). Interestingly, the relative agreement between mean duct area and $L_M$ was poorer for the samples from the BPD cohort when compared to normoxic controls, though this is likely attributable to variances in disease phenotype.

In keeping with existing literature [19,20], our method revealed that the average alveolar size was significantly increased following exposure to acute $O_2$ (Fig 4A,B). Additionally, the number of alveoli per FOV and the total area occupied by alveoli were significantly decreased in mice exposed to the hyperoxic model of BPD (Fig 4C,D). Although the association between mean alveolar size and $L_M$ remained significant (r: 0.5823, p = 0.0045), the association was weaker than what was computed on optimization data. No proportional bias was evident, and with the exception of two, the majority of samples remained within the 95% LOA (−1.791, 1.791; Fig 4F).

Quantification of intermediate airways, which could represent the terminal regions of alveolar ducts, alveolar sacs, or very large alveoli, revealed no significant differences in mean area (Fig 5A,B), suggesting this area range captures a mixed population. Despite equivalencies in mean area, the number of intermediate airspaces was significantly reduced in mice exposed to 95% $O_2$, alongside a reduction in the total area occupied by these airspaces (Fig 5C,D). Both alveoli and intermediate airspaces comprised a significantly reduced proportion of total airspace area in diseased mice, relative to controls (Fig 5E), with a corresponding increase in the proportion of total airspace area occupied by alveolar ducts being

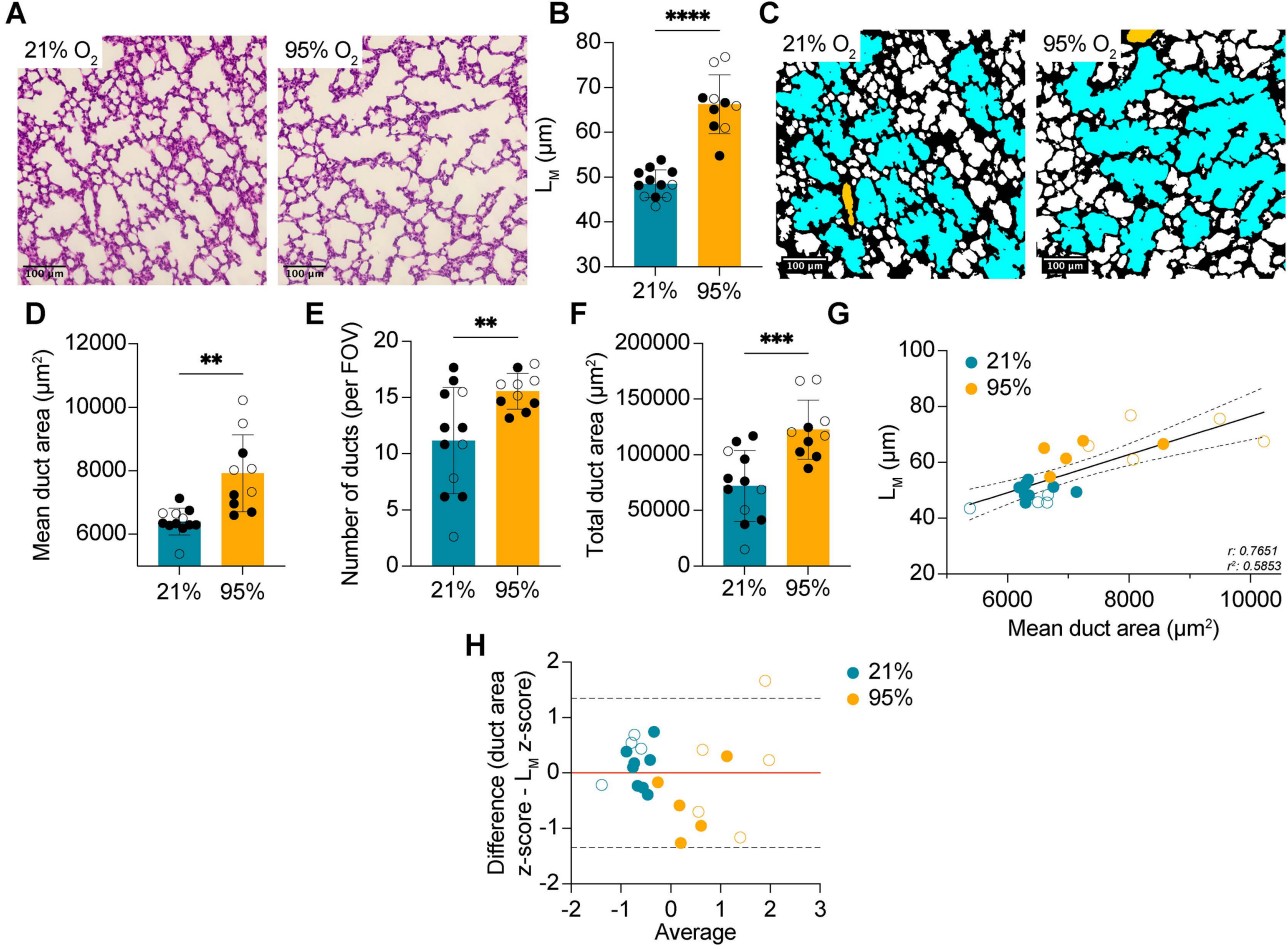

**Fig 3. Exposure to postnatal hyperoxia increases alveolar duct size and number.** (A) Representative H&E-stained images from mice exposed to 21% $O_2$ ($n=12$) or 95% $O_2$ ($n=10$) for 72 hours. (B) Quantification of $L_M$ in the mice from A. (C) Representative colored images depicting alveolar ducts. (D) Quantification of mean alveolar duct area from the same mice as A. (E) Quantification of the number of ducts per 20x FOV in the same mice as A. (F) Quantification of total alveolar duct area in the same mice as A. (G) Scatter plot of mean alveolar duct area plotted against measured $L_M$. Solid black line depicts the fitted linear regression line. Dashed lines represent the 95% confidence intervals. (H) Bland-Altman plot of z-scores for alveolar area and $L_M$. Dashed lines represent the 95% limit of agreement (LOA). Red line depicts bias. (○) female mice, (●) male mice. ****$P<0.0001$, ***$P<0.001$, P**$<0.01$. Student's t-test used in B,D,E,F. Simple linear regression used in G. Bland-Altman analysis used in H. Error bars are mean±SD.

observed. Taken together, these findings support the notion of impaired septation as a consequence of acute $O_2$ exposure during the early postnatal period.

## Discussion

Here, we demonstrate the utility of a semi-automated method to quantitate the characteristics of both alveolar ducts and alveoli in the distal lung. This method significantly reduces the time required to provide alveolar duct and alveolar counts, while removing the potential for operator bias. Consequently, we propose that this method can be utilized as a means of supporting spatial analyses involving distal lung tissue that use conventional manual measures. Although the optimized set of parameters were validated using tissues from BPD mice versus normoxia-exposed healthy controls, it is likely that this tool can be applied to tissues from other mouse models of lung disease.

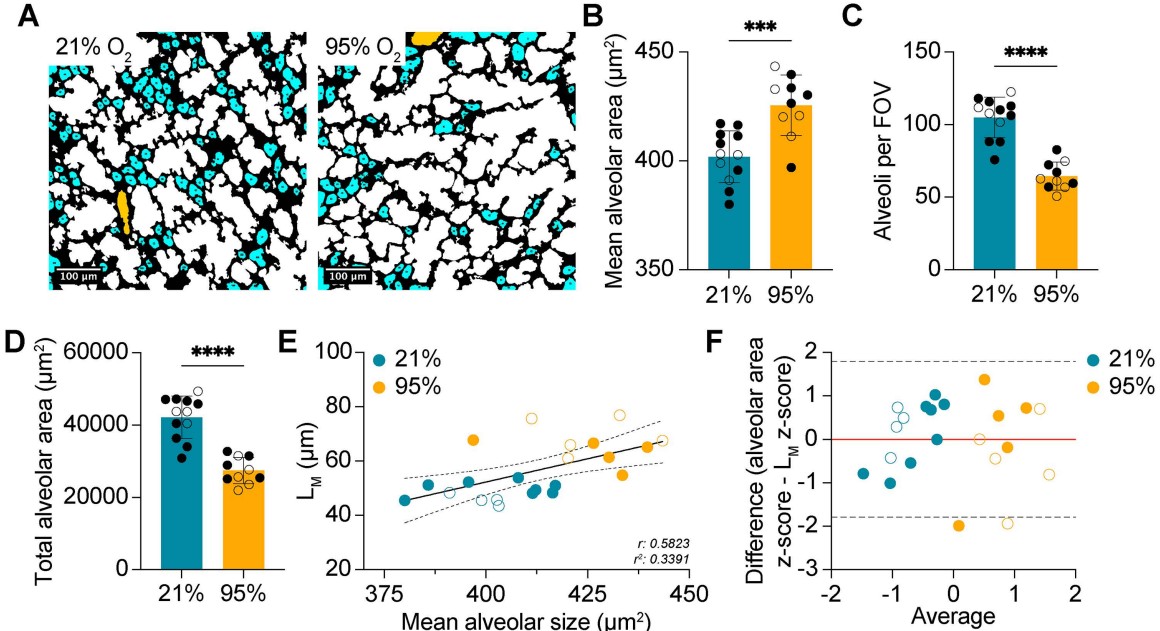

**Fig 4. Neonatal hyperoxia impairs alveolar septation.** (A) Representative colored images depicting alveoli from mice exposed to 21% $O_2$ (n = 12) or 95% $O_2$ (n = 10) for 72 hours. (B) Quantification of mean alveolar area from the same mice as A. (C) Quantification of the number of alveoli per 20x FOV in the same mice as A. (D) Quantification of total alveolar area in the same mice as A. (E) Scatter plot of mean alveolar duct area plotted against measured $L_M$. Solid black line depicts the fitted linear regression line. Dashed lines represent the 95% confidence intervals. (F) Bland-Altman plot of z-scores for alveolar area and $L_M$. Dashed lines represent the 95% limit of agreement (LOA). Red line depicts bias. (○) female mice, (●) male mice. ****$P < 0.0001$, ***$P < 0.001$, $P** < 0.01$. Student's t-test used in B,C,D. Simple linear regression used in E. Bland-Altman analysis used in F. Error bars are mean ± SD.

While the method is capable of differentiating between alveolar ducts and alveoli, this functionality is dependent on several important assumptions. Firstly, the tool assumes that all large terminal respiratory airways are alveolar ducts, and all small airways are alveoli. This assumption may not be explicitly true, and is dependent on the region of the lung being imaged, orientation, and several other factors that may result in sectioning through a small portion of a large airway. To avoid potential overlap between the two different generations of airways, we included a buffer zone for minimum area. This buffer zone was designed to limit the inclusion of both alveolar sacs and intermediate airways that could represent either of the two types. While this buffer may bias the resultant outputs towards slightly smaller alveoli and slightly larger alveolar ducts, the effect is likely minimal, and does not impede the capture of disease effects in our model. This rationale is supported by equivalencies in mean intermediate airspace area between BPD and control mice, despite differences in the proportion of total airway area being present. Additionally, by allowing for the removal of vascular and nontarget airway structures that may be included in other measurements, we increase the specificity of the results by providing outputs that arise solely from alveolar ducts and alveoli. This feature contrasts with other existing methods of airway assessment that provide related measures, but fail to exclude structures such as large vessels or focus on all airways in the lung, including conducting airways [16,21].

Inter-observer bias is a consideration for manual methods such as RAC or $L_M$. RAC typically involves the placement of a line that sits perpendicular to the center of a respiratory bronchiole. Following placement, the number of alveoli the line transects is counted. This process is repeated several times per lung, and values are averaged per sample. Typically, the main source of inter-observer bias relates to the placement and angle of the perpendicular line. Different observers may choose different angles relative to the pleural space, or different terminal respiratory bronchioles for line placement, a factor that may be influenced by the appearance of said respiratory bronchioles. Although more rigorous, $L_M$ measurements

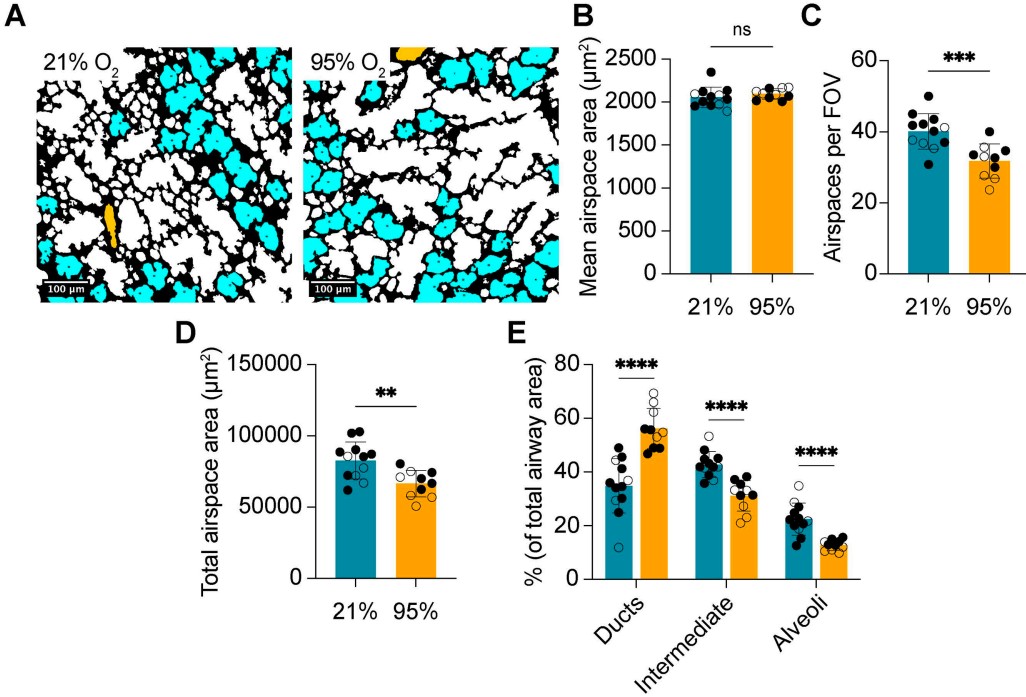

**Fig 5. Quantification of intermediate airspace characteristics on the validation dataset.** (A) Representative colored images depicting alveoli from mice exposed to 21% $O_2$ ($n = 12$) or 95% $O_2$ ($n = 10$) for 72 hours. (B) Quantification of mean airspace area from the same mice as A. (C) Quantification of the number of airspaces per 20x FOV in the same mice as A. (D) Quantification of total airspace area in the same mice as A. (E) Quantification of percent airway area for each subdivision of airspace in the same mice as A. Student's t-test used in B,C,D,E. Error bars are mean ± SD.

are also subjected to related biases, primarily relating to the definition of an intercept. Our method, which aims to supplement these existing measures, provides an unbiased approach to quantitating the characteristics of alveolar ducts and alveoli. In doing so, this tool allows for the determination of whether changes in RAC or $L_M$ are driven by changes at the level of one generation of airway, or both, in a manner that eliminates inter-observer bias. Furthermore, the highly automated nature of our method offers significant advantages when supplementing existing morphometric analyses, requiring little additional time or computational power.

While our method minimizes inter-observer bias, it cannot account for bias that results from heterogeneous lung processing (inflation, embedding orientation, sectioning), image acquisition, or the use of improper stereological processing techniques. However, this limitation applies to many of the manual morphometric analysis methods, including $L_M$ and RAC, and was not a focus for improvement in the development of our method. As this limitation is shared between analytical approaches, we postulate that our current method could be utilized independently to generate valid and valuable results. With the exception of stereology, methods of quantification, such as RAC and $L_M$, are typically performed concomitantly in order to provide a more comprehensive and reliable measure of lung morphology and mitigate some of the limitations discussed above. As such, we recommend that this macro be utilized in conjunction with existing measures to provide quantitate hierarchical specificity. However, it may be used independently if the operative assumptions are kept in mind.

In conclusion, we have provided an overview of the optimization and validation of a semi-automated macro to characterize alveolar ducts and alveoli in a mouse model of BPD. This method aims to diversify morphometric lung analyses by providing additional measures relating to specific classes of airways, offering insights into whether phenotypic changes

are driven by alterations in specific structures. With this method, we demonstrate that mice exposed to acute postnatal hyperoxia develop both enlarged alveoli that are less numerous and occupy a smaller total area, as well as enlarged alveolar ducts. The macro is freely available for use, and is appended as a .ijm macro file.

## Supporting information

**S1 File. Pearson correlation outputs for alveolar duct parameter optimization.**
(XLSX)

**S2 File. Pearson correlation outputs for alveolar parameter optimization.**
(XLSX)

**S3 File. Pearson correlation outputs for optimized test and validation set data.**
(XLSX)

**S4 File. Semi-automated FIJI macro.**
(TXT)

**S5 File. Raw data.**
(XLSX)

## Author contributions

**Conceptualization:** Declan J. Gainer, Mark L. Ormiston.

**Data curation:** Declan J. Gainer.

**Formal analysis:** Declan J. Gainer.

**Funding acquisition:** Mark L. Ormiston.

**Investigation:** Declan J. Gainer.

**Methodology:** Declan J. Gainer.

**Project administration:** Declan J. Gainer, Mark L. Ormiston.

**Software:** Declan J. Gainer.

**Supervision:** Mark L. Ormiston.

**Validation:** Declan J. Gainer.

**Visualization:** Declan J. Gainer.

**Writing – original draft:** Declan J. Gainer, Mark L. Ormiston.

**Writing – review & editing:** Declan J. Gainer, Mark L. Ormiston.

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
