## [Decision Letter · Decision Letter 0]

6 Nov 2025

Dear Dr.  Ormiston,

Thank you for submitting your manuscript to PLOS ONE. After careful consideration, we feel that it has merit but does not fully meet PLOS ONE’s publication criteria as it currently stands. Therefore, we invite you to submit a revised version of the manuscript that addresses the points raised during the review process.

Please submit your revised manuscript by Dec 21 2025 11:59PM. If you will need more time than this to complete your revisions, please reply to this message or contact the journal office at plosone@plos.org . A rebuttal letter that responds to each point raised by the academic editor and reviewer(s). You should upload this letter as a separate file labeled 'Response to Reviewers'.A marked-up copy of your manuscript that highlights changes made to the original version. You should upload this as a separate file labeled 'Revised Manuscript with Track Changes'.An unmarked version of your revised paper without tracked changes. You should upload this as a separate file labeled 'Manuscript'.

We look forward to receiving your revised manuscript.

Kind regards,

Yusuke Hoshino

Academic Editor

PLOS ONE

Journal Requirements:

Additional Editor Comments:

publication criteria

**Comments to the Author**

1. Is the manuscript technically sound, and do the data support the conclusions?

Reviewer #1: Yes

Reviewer #2: Yes

2. Has the statistical analysis been performed appropriately and rigorously?

Reviewer #1: Yes

Reviewer #2: Yes

3. Have the authors made all data underlying the findings in their manuscript fully available?

Reviewer #1: Yes

Reviewer #2: Yes

4. Is the manuscript presented in an intelligible fashion and written in standard English?

Reviewer #1: Yes

Reviewer #2: Yes

Reviewer #1: The work by Gainer and colleagues described a novel method to examine the distal patterning of the lung applied to models of bronchopulmonary dysplasia. The authors convey the importance of using unbiased methods to measure the distal lung architecture in lung disease. The manuscript describes a semi-automated method for quantifying alveolar ducts and alveoli using BPD as a model to test the tool. The work will be of interest to the journal's readership, particularly researchers in pulmonary biology. I am supportive of the manuscript's acceptance; however, the authors should address the following issues.

1- Authors should elaborate on whether the described method has the potential to be utilized independently or even replace standard measurements, such as the manual linear intercept.

2- While the validation using the BPD model is critical to validate the model, could the researchers perform additional measurements utilizing other disease models in older mice?

3- Could the macro be utilized or adapted to use with other image processing software?

4- Figure legend 1 is somewhat confused. For example, "(D) binarized version of D", or perhaps the binarized version of C. Please clarify.

Reviewer #2: This manuscript describes the development of a FIJI script to substantially automate the morphometric analysis of distal lung patterns in mammals. The semi-automated script reduces both hands-on time to make the measurements and possible operator bias. The developed script was benchmarked against the existing methods of mean linear intercept (LM) and was found to compare favorably.

The methods, statistics and analysis in the paper appear to be appropriate. The detailed statistical tests exceed this reviewer’s knowledge of statistics, but visual inspection of the processed and raw data suggests that the method is accurate and very useful. The text is generally clear and the diagrams are generally good, although specific suggestions to improve the text and figures are included below.

Overall this paper is appropriate to publish in PLOS ONE and should significantly benefit pulmonary research.

Concern:

1) In Figure 1 H and I, there is a considerable amount of tissue that is non marked off as non-targeted structures, and is neither large airways nor alveoli. (i.e. if you merge H and I, what is all the unmarked white space?). I don’t think this is described in the text, but I think there should be some explanation of the uncategorized tissue and some quantification of it. I would guess that at least 20% of the image is unaccounted-for space. Marking an example in Fig. 1I and explaining it in the text or legend would help a lot.

Suggestions to improve manuscript:

1) line 48 and elsewhere: The authors report their results show that hyperoxia-exposed mice possess fewer, enlarged alveoli that occupy less total area as well as enlarged alveolar ducts that occupy more space. How much of this was known before (which make for good validation of the technique) and how much is new results? If know, references are in order. If new results, a phase statement something along the lines of “ …. , effects that were not previously known.”

2) line 53: How much faster is using the pipeline than doing a manual LM? The manuscript claims improved processing times but does not quantify the improvements.

3) line 162. Why were a subset of mice given a subcutaneous dose of 5 ul of PBS prior to hyperoxic exposure. First, why was this done? There does not seem to be a treatment involving injection of a drug, so why inject such a small amount of PBS below the skin? Second, how was 5 ul of PBS reliably delivered? 5 ul in a microfuge tube is not hard, but 5ul subcutaneously seems technically challenging. Third, where was the mouse injected? Fourth, I couldn’t find a sub-group of injected broken out as a separate data set? Was this group pooled with the non-injected or was the data not used? If not used, please remove this description from methods.

4) lines 231 and 234 Tables 1 and 2. This needs a legend, and I think it would be less confusing if “Circularity” were replaced with “Circularity values tested”. There needs to some explaination for why there are two or three values in each cell on the Circularity row.

5) Figure 1. Lots of thing should be explicitly pointed out for the reader.

a. In C, there should be arrows or something indicating the manually removed nontarget structures. Making the black mark offs a bright orange would help make them visible.

b. in D and E, I could not actually find an example of a disconnected exudate or debris that had been removed. Please indicate in D and/or E examples of what changed.

c. in F/G I couldn’t find examples of small spaces that had been filled. If they are too small to see in the figure, include insets to show the effect.

d. Per the concern above, in H and I, there are still many spaces that are not colored in H or I. Why are these very nice looking spaces not colored and what percent of the total area do such spaces occupy? Please explain in the text or the legend.

6) Fig. 2 There are blue and orange circles is no key in the figure and no explanation in the legend of what these colors mean. Please add keys such as those in Fig. 3G or fig. 4E.

7) Fig. 2 The red dots overlapping with the blue dots in D are very hard to see. Can those be made a little bigger?

**Do you want your identity to be public for this peer review?** For information about this choice, including consent withdrawal, please see our Privacy Policy

Reviewer #1: No

Reviewer #2: No

---

## [Author Response · Author response to Decision Letter 1]

12 Jan 2026

Editor comments have been addressed. Style/naming conventions were changed to adhere with PLOS ONE's style requirements. Captions for supporting information files were added. Responses to specific reviewer comments were discussed in detain in the "Response to Reviews" document.

---

## [Editor Report · Decision Letter 1]

21 Jan 2026

An automated morphometric approach to evaluate distal lung patterning in mouse models of Bronchopulmonary Dysplasia

PONE-D-25-49190R1

Dear Dr. Mark L. Ormiston,

We’re pleased to inform you that your manuscript has been judged scientifically suitable for publication and will be formally accepted for publication once it meets all outstanding technical requirements.

Kind regards,

Yusuke Hoshino

Academic Editor

PLOS One
---

## [Editor Report · Acceptance letter]

PONE-D-25-49190R1

PLOS One

Dear Dr. Ormiston,

I'm pleased to inform you that your manuscript has been deemed suitable for publication in PLOS One. Congratulations! Your manuscript is now being handed over to our production team.

Kind regards,

on behalf of

Dr. Yusuke Hoshino

Academic Editor

PLOS One